# Which Out-of-Hospital Cardiac Arrest Patients without ST-Segment Elevation Benefit from Early Coronary Angiography? Results from the Korean Hypothermia Network Prospective Registry

**DOI:** 10.3390/jcm10030439

**Published:** 2021-01-23

**Authors:** Hwan Song, Hyo Joon Kim, Kyu Nam Park, Soo Hyun Kim, Won Young Kim, Byung Kook Lee, In Soo Cho, Jae Hoon Lee, Chun Song Youn

**Affiliations:** 1Department of Emergency Medicine, Seoul St. Mary Hospital, College of Medicine, The Catholic University of Korea, Seoul 06591, Korea; cmcmdsong@gmail.com (H.S.); liebestest@hanmail.net (H.J.K.); emsky@catholic.ac.kr (K.N.P.); 2Department of Emergency Medicine, Eunpyeong St. Mary Hospital, College of Medicine, The Catholic University of Korea, Seoul 03312, Korea; unidgirl@catholic.ac.kr; 3Asan Medical Center, Department of Emergency Medicine, Ulsan University College of Medicine, Seoul 05505, Korea; wonpia73@naver.com; 4Department of Emergency Medicine, Chonnam National University Medical School, Gwangju 61469, Korea; bbukkuk@hanmail.net; 5Department of Emergency Medicine, Hanil General Hospital, Korea Electric Power Medical Corporation, Seoul 01450, Korea; mensa@hanmail.net; 6Department of Emergency Medicine, Dong-A University College of Medicine, Busan 49201, Korea; leetoloc@dau.ac.kr

**Keywords:** out-of-hospital cardiac arrest, cardiopulmonary resuscitation, coronary angiography, outcome

## Abstract

The effect of early coronary angiography (CAG) in out-of-hospital cardiac arrest (OHCA) patients without ST-elevation (STE) is still controversial. It is not known which subgroups of patients without STE are the most likely to benefit. The objective of this study was to evaluate the association between emergency CAG and neurologic outcomes and identify subgroups with improved outcomes when emergency CAG was performed. This prospective, multicenter, observational cohort study was based on data from the Korean Hypothermia Network prospective registry (KORHN-PRO) 1.0. Adult OHCA patients who were treated with targeted temperature management (TTM) without any obvious extracardiac cause were included. Patients were dichotomized into early CAG (≤24 h) and no early CAG (>24 h or not performed) groups. High-risk patients were defined as having the Global Registry of Acute Coronary Events (GRACE) score > 140, time from collapse to return of spontaneous circulation (ROSC) > 30 min, lactate level > 7.0 mmol/L, arterial pH < 7.2, cardiac enzyme elevation and ST deviation. The primary outcome was good neurologic outcome at 6 months after OHCA. Of the 1373 patients from the KORHN-PRO 1.0 database, 678 patients met the inclusion criteria. The early CAG group showed better neurologic outcomes at 6 months after cardiac arrest (CA) (adjusted odds ratio: 2.21 (1.27–3.87), *p* = 0.005). This was maintained even after propensity score matching (adjusted odds ratio: 2.23 (1.39–3.58), *p* < 0.001). In the subgroup analysis, high-risk patients showed a greater benefit from early CAG. In contrast, no significant association was found in low-risk patients. Early CAG was associated with good neurologic outcome at 6 months after CA and should be considered in high-risk patients.

## 1. Introduction

The survival rate after out-of-hospital cardiac arrest (OHCA) remains low despite recent advances in critical care [1,2]. Considering that coronary artery disease is the most common cause of OHCA, early coronary reperfusion may improve outcomes after OHCA [3,4,5]. Current guidelines highlight the role of emergency coronary angiography (CAG) and percutaneous coronary intervention (PCI) for OHCA patients with ST-segment elevation (STE) on electrocardiogram (ECG) after the return of spontaneous circulation (ROSC) [6,7]. However, the effect of emergency CAG on OHCA patients without STE is still controversial [6,8,9].

Some recent studies have found coronary occlusions in approximately 30% of patients without STE [10], and early coronary revascularization may increase the survival rate through hemodynamic improvement. Moreover, improvement in hemodynamic status improves heart function and brain blood flow, which can improve neurological outcomes [11]. However, implementing this invasive strategy in routine clinical practice has several logistical and organizational problems. Furthermore, CAG could delay targeted temperature management (TTM), which is standard care after OHCA, and could delay the appropriate diagnosis of other possible causes of cardiac arrest (CA) [11,12,13,14,15]. Observational studies have shown conflicting results regarding the effect of early CAG on the outcome of patients without STE [9,16,17,18,19]. In a recent randomized controlled trial (the coronary angiography after cardiac arrest (COACT) trial), immediate angiography was not superior to delayed angiography among patients who had no signs of STE myocardial infarction after ROSC [20]. However, the COACT trial only includes patients with a shockable rhythm, making it difficult to generalize these results to all cardiac arrest patients. Despite limited evidence, recent postcardiac arrest care guidelines recommend early CAG in selected patients after ROSC, even for patients without STE [7,21,22]. Surprisingly, no study has investigated the selection of candidates for emergency CAG based on patient data available at the time of hospital admission. Therefore, it is important to identify subgroups in which emergency CAG in patients without STE can lead to improved prognosis.

The purpose of this study was to examine the association between emergency CAG and outcomes in patients without STE after OHCA and to identify subgroups with improved outcomes when emergency CAG was performed.

## 2. Methods

### 2.1. Study Design and Setting

This was a prospective, multicenter, observational cohort study. Data were collected from the Korean Hypothermia Network prospective registry (KORHN-PRO) 1.0 between October 2015 and December 2018. KORHN is a multicenter clinical research consortium for TTM in South Korea. Twenty-two academic hospitals participated in the KORHN-PRO. The study included an informed consent form approved by all participating hospitals, including the institutional review board (IRB) of Seoul St. Mary’s Hospital (XC15OIMI0081K), and the study was registered at the International Clinical Trials Registry Platform (NCT02827422). Written informed consent was obtained from all patients′ legal surrogates.

### 2.2. Population

The inclusion criteria of KORHN-PRO were as follows: OHCA regardless of etiology of cardiac arrest, age older than 18 years, unconsciousness (Glasgow Coma Scale score < 8) after ROSC and treatment with TTM. The exclusion criteria were as follows: active intracranial bleeding, acute stroke, known limitations in therapy and a do-not-attempt resuscitation order, known prearrest cerebral performance category (CPC) 3 or 4, known disease making 6-month survival unlikely, and body temperature < 30 °C on admission.

For the present trial, we excluded patients who had obvious non-cardiac causes (e.g., hanging, asphyxia, drowning, etc.) and STE on the initial ECG from KORHN-PRO. In addition, we included shockable rhythm patients as well as non-shockable rhythm patients in the study, which is different from the COACT trial.

### 2.3. Variables

The primary outcome was good neurologic outcome at 6 months. The cerebral performance category (CPC) was used to assess neurologic outcome, with a CPC score of 1–2 considered a good outcome and a CPC of 3–5 a poor outcome.

We collected the following demographic and clinical variables information from the registry: age, sex, comorbidities (previous arrest, previous acute myocardial infarction, previous PCI, previous coronary artery bypass grafting (CABG), angina, arrhythmia, chronic heart failure, cerebrovascular accident (CVA) or transient ischemic attack (TIA), hypertension, diabetes, pulmonary disease, chronic kidney disease, liver cirrhosis, malignancy, smoking, alcohol), resuscitation variables (witnessed arrest, bystander cardiopulmonary resuscitation (CPR), initial rhythm, time from collapse to ROSC), and post-ROSC variables (ST deviation, cardiac enzyme elevation, arterial blood gas analysis, lactate level). The Global Registry of Acute Coronary Events (GRACE) risk score was calculated based on the information from the registry.

### 2.4. Definitions

We defined “early CAG” as performed within 24 h after ROSC [11,18]. “No early CAG” was defined as CAG performed after 24 h or not performed at all. ECG patterns were classified as follows: normal, ST depression, left bundle branch block (LBBB), right bundle branch block (RBBB), nonspecific ST or T change. ST depression and nonspecific ST or T change were included in ST deviation. The upper limits of cardiac enzyme levels for troponin-I and troponin-T were defined as 3 ng/dL and 1.4 ng/dL, respectively.

High-risk patients were defined by a GRACE score > 140, time from collapse to ROSC > 30 min, lactate level > 7.0 mmol/L and arterial pH < 7.2 [22,23,24].

### 2.5. Statistical Methods

Normality tests (Shapiro–Wilk test) were performed for continuous variables, and continuous variables are presented as the means with the standard deviation (mean ± sd) or as median values with interquartile ranges, as appropriate. Categorical variables are presented as frequencies and percentages. For patient characteristics and comparisons between groups, we used Student’s t-test or the Mann–Whitney U test for continuous variables and Fisher’s exact test and the chi-square test for categorical variables.

The baseline characteristics of our cohorts are not balanced. Thus, 1:1 or 1:M matching was applied for balanced covariates. By comparison group (early CAG and non-early CAG), we calculated the propensity score (which was calculated using logistic regression). After this analysis, we performed the 1:1 and 1:2 matching by nearest method and checked the standardized difference (*d*). As a result, we used 1 (early CAG):2 (non-early CAG) matching data for this study.

After propensity score matching, univariate analysis was performed to determine the covariates for neurologic outcome at 6 months after CA. Variables with a *p*-value ≥ 0.05 on univariate analysis were excluded from the multivariate logistic regression model. To examine the association between early CAG and good neurologic outcome at 6 months after CA, multivariate logistic regression analyses with backward elimination were performed. We then performed multivariate logistic regression analysis among each subgroup of high-risk patients (GRACE score > 140, time from collapse to ROSC > 30 min, lactate level > 7.0 mmol/L and arterial pH < 7.2) and low-risk patients. The association between early CAG and outcome was quantified using odds ratios (ORs) with 95% confidence intervals (CIs).

Statistical analysis was performed using SAS version 9.4 (SAS Institute, Inc., Cary, NC, USA) and R version 4.0.3, and *p*-values ≤ 0.05 were considered statistically significant.

## 3. Results

### 3.1. Study Population

During the study period, a total of 1373 OHCA patients older than 18 years who were treated with TTM were enrolled in KORHN-PRO. Among them, 521 patients were excluded due to obvious non-cardiac causes of arrest, and 11 patients were excluded due to lack of ECG data. STE was found in 163 patients, and the remaining 678 patients were ultimately included in this study (Figure 1). CAG was performed within 24 h in 231 patients (early CAG group), and the remaining 447 patients either underwent CAG after 24 h or were not treated with CAG (no early CAG group).

The mean age was not significantly different between the early CAG group (58.5 ± 13.8) and the no early CAG group (59.6 ± 16.1, *p* = 0.357), and 497 (73.3%) patients were male. There were no significant differences between the early CAG group and the no early CAG group in comorbidities, except chronic heart failure and pulmonary disease. The proportion of good neurologic outcome at 6 months after CA was higher in the early CAG group than in the no early CAG group (51.1% vs. 37.6%, *p* < 0.001).

An initial shockable rhythm was more common in the early CAG group than in the no early CAG group (68.3% vs. 47.9%, *p* < 0.001). Moreover, cardiac enzyme elevation was also more common in the early CAG group than in the no early CAG group (83.9% vs. 72.0%, *p* < 0.001). The time from collapse to ROSC and GRACE score showed no significant differences between the two groups (Table 1).

### 3.2. ECG and CAG Findings

ST depression was more common in the early CAG group (102 (44.2%) vs. 106 (23.7%), *p* < 0.001), while nonspecific ST or T was more common in the no early CAG group (212 (47.4%) vs. 67 (29.0%), *p* < 0.001).

The median time from ROSC to CAG was 2.2 h (interquartile range (IQR) 1.5–3.3 h) in the early CAG group and 176.7 h (IQR 12.6–229.5) in the no early CAG group. Coronary artery disease, defined as more than 50% stenosis of any coronary artery, was 41.1% in the early CAG group, and the left anterior descending artery was the most commonly involved artery (66.3%). Notably, CAG was performed in 104 (23.3%) patients in the no early CAG group, and only 26 (5.8%) patients required subsequent PCI. In contrast, 76 (32.9%) patients in the early CAG group underwent PCI (Table 2).

### 3.3. Association between Early CAG and Outcomes in the Whole Population

In the univariate analysis, age, previous PCI, history of high blood pressure (HBP), diabetes mellitus (DM), pulmonary disease and chronic kidney disease, witnessed arrest, bystander CPR, initial shockable rhythm, time from collapse to ROSC > 30 min, arterial pH < 7.2, lactate > 7.0 mmol/L and GRACE score > 140 were associated with good neurologic outcome at 6 months after CA.

In the multivariate logistic regression analysis, the early CAG group was associated with good neurologic outcome at 6 months after CA (adjusted OR: 2.21 (1.27–3.87), *p* = 0.005) (Table 3). The cardiovascular Sequential Organ Failure Assessment (SOFA) score from the third to seventh day after ROSC differed between the early CAG group and the no early CAG group in patients with a GRACE score of 140 or higher immediately after ROSC. However, in the group with a GRACE score ≤ 140, the cardiovascular SOFA score showed no significant difference between the two groups (Figure 2).

After propensity score matching, early CAG was associated with good neurologic outcome at 6 months after CA (adjusted OR: 2.23 (1.39–3.58), *p* < 0.001) (Table 3).

### 3.4. Impact of Early CAG in Selected High-Risk Patients

We performed subgroup analysis between the early CAG group and the no early CAG group to assess good neurologic outcomes at 6 months. Early CAG was associated with good neurologic outcome at 6 months after CA in high-risk patients (defined as GRACE score > 140, time from collapse to ROSC > 30 min, arterial pH < 7.2, lactate > 7.0 mmol/L, cardiac enzyme elevation, and ST deviation). In contrast, no significant association was found among patients with GRACE score ≤ 140, time from collapse to ROSC ≤ 30 min, lactate level ≤ 7.0 mmol/L, arterial pH ≥ 7.2, no cardiac enzyme elevation and no ST deviation (Table 4).

## 4. Discussion

The main findings of this study are as follows: first, early CAG was associated with improved long-term neurologic outcome at 6 months after CA in patients treated with TTM. Second, we demonstrated an association between early CAG and improved long-term neurologic outcome among high-risk patients (i.e., GRACE score > 140, time from collapse to ROSC > 30 min, lactate level > 7.0 mmol/L, arterial pH < 7.2, cardiac enzyme elevation and ST deviation). In contrast, no significant association was found among patients with GRACE score ≤ 140, time from collapse to ROSC ≤ 30 min, lactate level ≤ 7.0 mmol/L, arterial pH ≥ 7.2, no cardiac enzyme elevation and no ST deviation.

Early CAG has the advantage of early implementation of PCI, which can most effectively improve the hemodynamic status of patients with coronary artery disease (CAD), ameliorating the cerebral perfusion of patients and resulting in an improvement in long-term neurologic outcomes. We demonstrated that the cardiovascular SOFA score from the third to the seventh day after ROSC differed between the early CAG group and the no early CAG group in patients with a GRACE score of 140 or higher immediately after ROSC. However, there is no reason for emergency CAG in patients whose neurological outcome is expected to be futile. If there is no STE on the electrocardiogram after CA, there are two important issues when deciding whether to perform emergency CAG: right patients and right timing.

The optimal timing of CAG after CA is still unknown. Kim et al. stated that immediate CAG within 2 h after ROSC or emergency department presentation had no clear neurological benefit compared with early CAG between 2 h and 24 h [3]. In addition, several retrospective studies have defined early CAG as being performed within 24 h after ROSC [3,11,25]. Based on this, we classified patients who underwent CAG within 24 h as the early CAG group [23]. However, we did not find any evidence for correct timing in this study. Future studies are needed.

According to previous systematic reviews, CAD was diagnosed in one-third of patients who underwent CAG without STE [10,26]. In our study, the early CAG group was diagnosed with CAD in 41.1% of patients, similar to previous systematic reviews. However, in the no early CAG group, CAD was diagnosed in 7.2% of the patients, and among the no early CAG group, CAD was diagnosed in 30.8% of the patients who underwent CAG; thus, CAD was diagnosed among patients who underwent CAG at a rate similar to that of previous studies.

Comatose patients after CA with evidence of STE should undergo emergency CAG. However, it is challenging to decide which patients should undergo emergency CAG without evidence of STE, i.e., the “right” patients. Therefore, we defined high-risk patients based on the American Heart Association/American College of Cardiology (AHA/ACC) guidelines and the algorithm proposed by Rab et al. [22,23]. For non-cardiac arrest patients, CAG is recommended within 24 h in acute coronary syndrome patients without STE with a GRACE score > 140 [27]. Rab et al. proposed an algorithm for the risk stratification of comatose CA patients. Time from collapse to ROSC of 30 min or more, arterial pH < 7.2, and lactate level > 7 mmol/L were included in the unfavorable resuscitation features. However, our results presented in Table 4 show that the benefit of early CAG was greater in high-risk patients. In contrast, low-risk patients did not significantly benefit from early CAG. Our research does not contradict the algorithm from Rab et al.; however, an improvement in cardiovascular SOFA scores by emergency CAG was found for high-risk patients, suggesting the possibility of improving their neurological prognosis.

The COACT trial, which is a recently published randomized controlled trial (RCT) for OHCA patients without STE, showed no significant differences in survival at 90 days between immediate CAG and delayed CAG [20]. A possible explanation for the difference between the COACT trial and our study might be the baseline characteristics of the enrolled patients. The COACT trial only included patients with shockable rhythm, which made up half of the patients in our study. Patients who received immediate CAG in the COACT trial had a shorter down time (15 min vs. 32 min), higher levels of arterial pH (7.2 vs. 7.13), and lower levels of lactate on admission (4.9 mmol/L vs. 9.15 mmol/L) than those of the patients in our study. High-risk patients were more likely to be included in our study than in the COACT trial. Therefore, our study can provide additional information on patients not included in the COACT trial. Although observational studies may require additional RCTs and large-scale studies, our results suggest that early CAG should be performed in high-risk groups.

There were several limitations in this study. Our study is an observational prospective study and may lead to the risk of selection bias and residual confounding. An important possible explanation for the difference between the COACT trial and our study may be due to selection bias. Although the baseline characteristics, as well as the GRACE score, were not significantly different between the early CAG group and the no early CAG group in our study, the decision-making process of early CAG for each institution was inconsistent, and the possibility of CAG being performed earlier in patients with a predicted good prognosis cannot be excluded. CAG was not performed in 76.7% of the no early CAG group, which may have affected the outcome. However, the proportion of patients with PCI among the patients who underwent CAG was similar in both groups.

## 5. Conclusions

In conclusion, CAG performed within 24 h is associated with good neurologic outcome at 6 months after CA in patients without STE on ECG with presumed cardiac etiology treated with TTM. In particular, early CAG may be beneficial for high-risk patients (i.e., GRACE score > 140, time from collapse to ROSC > 30 min, cardiac enzyme elevation, arterial pH < 7.2, lactate > 7.0 mmol/L, cardiac enzyme elevation and ST deviation).

## Figures and Tables

**Figure 1 jcm-10-00439-f001:**
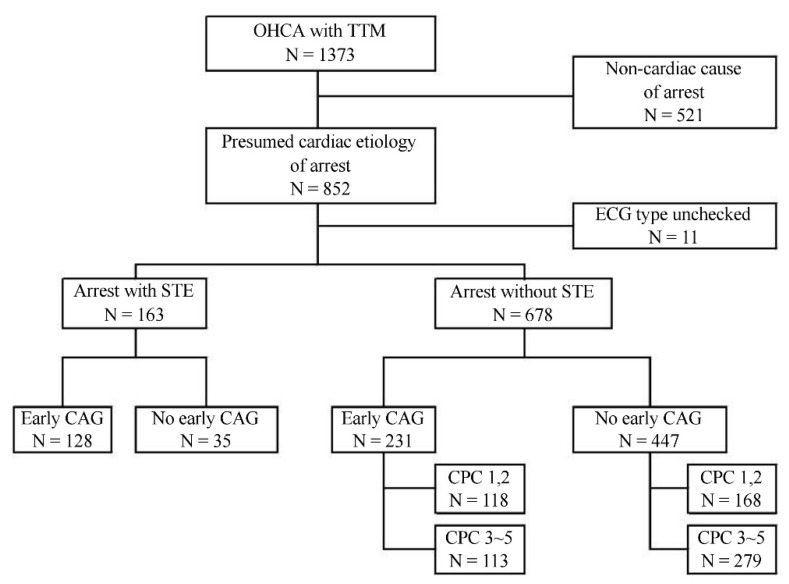
Flow chart of patients. Abbreviations: OHCA = out-of-hospital cardiac arrest, TTM = targeted temperature management, ECG = electrocardiogram, STE = ST segment elevation, CAG = coronary angiography, CPC = cerebral performance category.

**Figure 2 jcm-10-00439-f002:**
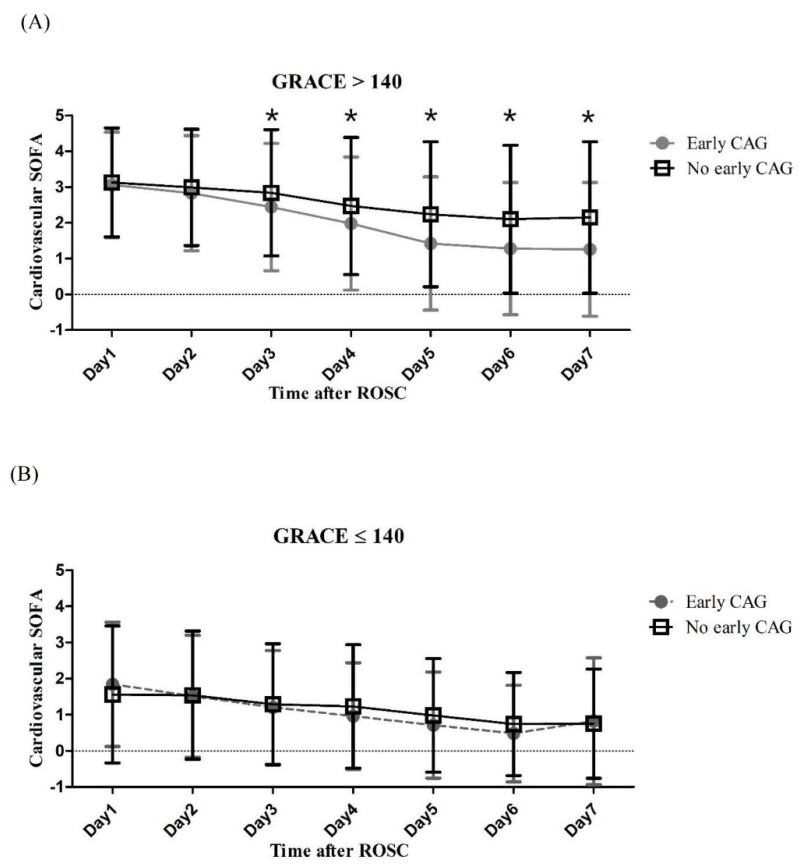
Cardiovascular SOFA score within 7 days after ROSC according to GRACE score. (**A**) Cardiovascular SOFA score in patients with GRACE score > 140 (**B**) Cardiovascular SOFA score in patients with GRACE score ≤ 140. * Indicates a *p* value of less than 0.05.

**Table 1 jcm-10-00439-t001:** Baseline characteristics of patients with out-of-hospital cardiac arrest without ST-segment elevation.

	Before Propensity Score Matching			After Propensity Score Matching		
	No Early CAG (N = 447)	Early CAG (N = 231)	*p* Value	No Early CAG (N = 364)	Early CAG (N = 182)	*p* Value
Age, years	59.61 ± 16.08	58.52 ± 13.75	0.357	58.76 ± 15.90	58.26 ± 14.05	0.399
Sex, male	317 (70.9)	180 (77.9)	0.051	262 (72.0)	139 (76.4)	0.320
Comorbidities						
Previous arrest	10 (2.2)	2 (0.9)	0.237	7 (1.9)	2 (1.1)	0.725
Previous AMI	31 (6.9)	24 (10.4)	0.118	24 (6.6)	16 (8.8)	0.450
Previous PCI	18 (4.0)	15 (6.5)	0.157	16 (4.4)	12 (6.6)	0.373
Previous CABG	6 (1.3)	5 (2.2)	0.523	4 (1.1)	3 (1.6)	0.691
Angina	31 (6.9)	24 (10.4)	0.118	21 (5.8)	15 (8.2)	0.360
Arrhythmia	32 (7.2)	15 (6.5)	0.747	29 (8.0)	14 (7.7)	1.000
Chronic heart failure	28 (6.3)	6 (2.6)	**0.038**	19 (5.2)	6 (3.3)	0.426
CVA or TIA	27 (6.0)	9 (3.9)	0.238	15 (4.1)	5 (2.7)	0.573
Hypertension	189 (42.3)	86 (37.2)	0.204	145 (39.8)	71 (39.0)	0.926
Diabetes mellitus	118 (26.4)	56 (24.2)	0.543	90 (24.7)	42 (23.1)	0.750
Pulmonary disease	30 (6.7)	2 (0.9)	**0.001**	14 (3.8)	2 (1.1)	0.127
Chronic kidney disease	41 (9.2)	12 (5.2)	0.068	23 (6.3)	8 (4.4)	0.472
Liver cirrhosis	7 (1.6)	0 (0.0)	0.102	0 (0.0)	0 (0.0)	1.000
Malignancy	27 (6.0)	10 (4.3)	0.353	19 (5.2)	9 (4.9)	1.000
Smoking	111 (25.5)	80 (36.0)	**0.005**	62 (22.6)	49 (32.7)	**0.025**
Alcohol	144 (33.0)	92 (41.4)	**0.033**	91 (33.2)	64 (42.7)	0.053
Resuscitation variables						
Witness	354 (79.6)	184 (80.0)	0.891	292 (80.2)	144 (79.1)	0.850
Bystander	281 (63.6)	154 (67.5)	0.308	235 (64.6)	127 (69.8)	0.263
Shockable	201 (47.9)	138 (68.3)	**<0.001**	186 (51.1)	125 (68.7)	**<0.001**
Time from collapse to ROSC, min	31.77 ± 22.38	32.08 ± 21.37	0.862	31.65 ± 24.08	32.72 ± 20.59	0.248
Time from collapse to ROSC > 30 min	200 (44.7)	100 (43.3)	0.718	158 (43.4)	83 (45.6)	0.692
Post-resuscitation variables						
ST deviation	303 (67.8)	167 (72.3)	0.228	247 (67.9)	129 (70.9)	0.535
Cardiac enzyme elevation	321 (72.0)	193 (83.9)	**0.001**	263 (72.3)	150 (82.4)	**0.012**
Arterial pH	7.09 ± 0.21	7.13 ± 0.18	**0.023**	7.10 ± 0.21	7.13 ± 0.18	0.117
Arterial pH < 7.2	277 (63.8)	126 (57.5)	0.118	223 (61.3)	104 (57.1)	0.405
Lactate, mmol/L	9.09 ± 4.92	9.15 ± 4.85	0.884	8.99 ± 5.07	9.12 ± 5.04	0.885
Lactate > 7 mmol/L	267 (62.1)	142 (64.3)	0.589	228 (62.6)	117 (64.3)	0.778
GRACE score	179.85 ± 52.74	178.23 ± 48.58	0.696	177.45 ± 54.51	174.74 ± 47.97	0.524
GRACE score > 140	337 (75.4)	183 (79.2)	0.264	269 (73.9)	143 (78.6)	0.276
TTM variables						
Target temperature of 33 °C	381 (85.2)	168 (72.7)	**<0.001**	251 (89.6)	119 (76.3)	**<0.001**
Target temperature of 36 °C	66 (14.8)	63 (27.3)	**<0.001**	29 (10.4)	37 (23.7)	**<0.001**
Time to achieve target temperature, min	424.60 ± 277.20	484.90 ± 290.87	**0.009**	389.47 ± 239.36	490.44 ± 313.89	**<0.001**
Survival discharge	255 (57.0)	163 (70.6)	**0.001**	164 (58.6)	112 (71.8)	**0.006**
Survival at 6 months	207 (47.4)	131 (58.2)	**0.008**	131 (47.3)	93 (60.4)	**0.009**
Good neurologic outcome at 6 months	168 (37.6%)	118 (51.1%)	**0.001**	107 (38.6)	82 (53.3)	**0.003**

*p* < 0.05 are presented in bold. Abbreviations: AMI = acute myocardial infarction, CVA = cerebrovascular accident, TIA = transient ischemic attack, HBP = hypertension, DM = diabetes mellitus, ABGA = arterial blood gas analysis, GRACE = Global Registry of Acute Coronary Events, ROSC: return of spontaneous circulation, PCI: percutaneous coronary intervention, CABG: coronary artery bypass grafting, TTM: targeted temperature management.

**Table 2 jcm-10-00439-t002:** Coronary angiography and electrocardiogram (ECG) findings.

	No Early CAG (N = 447)	Early CAG (N = 231)	*p* Value
Time to CAG, hours, median (IQR)	176.66 (125.68, 229.52)	2.2 (1.54, 3.33)	**<0.001**
No CAG	343 (76.7%)	N/A	
CAD (> =50%)	32 (7.2%)	95 (41.1%)	**<0.001**
1VD	18/32 (56.3%)	53/95 (55.8%)	0.964
2VD	8/32 (25.0%)	19/95 (20.0%)	0.550
3VD	6/32 (18.8%)	21/95 (22.1%)	0.688
LAD	23/32 (71.9%)	63/95 (66.3%)	0.561
LCx	12/32 (37.5%)	49/95 (51.6%)	0.168
RCA	17/32 (53.1%)	42/95 (44.2%)	0.382
Left main	0/32 (0%)	7/95 (7.4%)	0.190
Coronary intervention	26 (5.8%)	76 (32.9%)	**<0.001**
ST depression	106 (23.7%)	102 (44.2%)	**<0.001**
LBBB	18 (4.0%)	10 (4.36%)	0.851
RBBB	101 (22.6%)	50 (21.6%)	0.778
Non-specific ST or T	212 (47.4%)	67 (29.0%)	**<0.001**
Normal ST segment and T	76 (17.0%)	27 (11.7%)	0.068

*p* < 0.05 are presented in bold. Abbreviations; CAG = coronary angiogram, CAD = coronary artery disease, VD = vessel disease, LAD = left anterior descending artery, LCx = left circumflex artery, RCA = right coronary artery, LBBB = left bundle branch block, RBBB = right bundle branch block.

**Table 3 jcm-10-00439-t003:** Association between 6-month good neurologic outcome and baseline variables.

	Before Propensity Score Matching				After Propensity Score Matching			
	Crude		Adjusted		Crude		Adjusted	
	OR (95% CI)	*p* Value	OR (95% CI)	*p* Value	OR (95% CI)	*p* Value	OR (95% CI)	*p* Value
Early CAG	1.73 (1.26–2.39)	**0.001**	2.21 (1.27–3.87)	**0.005**	1.83 (1.25–2.67)	**0.002**	2.23 (1.39–3.58)	**<0.001**
Age, years	0.95 (0.94–0.96)	**<0.001**			0.95 (0.94–0.96)	**<0.001**	0.98 (0.96–1.00)	**0.011**
Sex (male)	0.72 (0.51–1.03)	0.069			1.35 (0.89–2.02)	0.155		
GRACE score (>140)	0.08 (0.05–0.13)	**<0.001**	0.15 (0.07–0.30)	**<0.001**	0.08 (0.05–0.13)	**<0.001**	0.24 (0.13–0.47)	**<0.001**
**Comorbidities**								
Previous arrest	1.02 (0.32–3.25)	0.971			0.61 (0.16–2.25)	0.454		
Previous AMI	1.55 (0.86–2.79)	0.141			0.48 (0.24–0.96)	**0.037**	1.16 (0.52–2.57)	0.716
Previous PCI	2.83 (1.21–6.62)	**0.016**	0.76 (0.21–2.78)	0.675	0.42 (0.17–1.05)	0.064		
Previous CABG	1.96 (0.52–7.47)	0.322			0.34 (0.07–1.78)	0.202		
Angina	0.80 (0.46–1.39)	0.426			1.10 (0.55–2.17)	0.795		
Arrhythmia	1.08 (0.59–1.98)	0.802			0.67 (0.34–1.32)	0.249		
Chronic heart failure	1.80 (0.85–3.83)	0.127			0.42 (0.17–1.04)	0.061		
CVA or TIA	1.70 (0.82–3.52)	0.151			0.87 (0.32–2.33)	0.777		
HBP	2.29 (1.66–3.15)	**<0.001**	0.78 (0.44–1.36)	0.376	0.47 (0.32–0.68)	**<0.001**	0.62 (0.39–1.00)	0.052
DM	3.33 (2.24–4.95)	**<0.001**	0.57 (0.28–1.15)	0.115	0.34 (0.21–0.54)	**<0.001**	0.77 (0.44–1.37)	0.376
Pulmonary disease	2.72 (1.16–6.37)	**0.022**	2.12 (0.64–7.04)	0.222	0.30 (0.06–1.47)	0.137		
Chronic kidney disease	3.00 (1.52–5.94)	**0.002**	0.31 (0.09–1.03)	0.056	0.32 (0.13–0.84)	**0.020**	0.30 (0.11–0.84)	**0.021**
malignancy	1.07 (0.55–2.11)	0.836			1.28 (0.58–2.84)	0.548		
Smoking	1.28 (0.92–1.80)	0.145			1.00 (0.66–1.52)	0.985		
Alcohol	1.97 (1.43–2.71)	**<0.001**	1.16 (0.67–2.00)	0.597	1.57 (1.08–2.30)	**0.020**	0.86 (0.54–1.37)	0.535
**Resuscitation variables**								
Witness	2.53 (1.66–3.85)	**<0.001**	1.87 (0.93–3.79)	0.080	2.87 (1.71–4.80)	**<0.001**	1.74 (0.98–3.07)	0.059
Bystander CPR	1.42 (1.03–1.97)	**0.035**	1.11 (0.63–1.96)	0.716	1.46 (0.99–2.13)	0.055		
Shockable rhythm	10.73 (7.23–15.92)	**<0.001**	6.67 (3.76–11.82)	**<0.001**	9.88 (6.41–15.22)	**<0.001**	3.94 (2.47–6.29)	**<0.001**
ROSC > 30 min	0.11 (0.08–0.16)	**<0.001**	0.16 (0.09–0.28)	**<0.001**	0.21 (0.08–0.18)	**<0.001**	0.18 (0.11–0.27)	**<0.001**
**Post-resuscitation variables**								
ST deviation	0.77 (0.55–1.07)	0.119			0.89 (0.60–1.32)	0.564		
Cardiac enzyme elevation	0.81 (0.57–1.15)	0.239			0.71 (0.47–1.07)	0.097		
Arterial pH < 7.2	0.20 (0.14–0.28)	**<0.001**	0.48 (0.27–0.85)	**0.012**	0.21 (0.14–0.31)	**<0.001**	0.44 (0.27–0.71)	**0.001**
Lactate > 7.0 mmol/L	0.31 (0.23–0.44)	**<0.001**	0.70 (0.39–1.24)	0.221	0.36 (0.25–0.53)	**<0.001**	0.72 (0.44–1.18)	0.191
**TTM variables**								
Target temperature of 33 °C	1.08 (0.73–1.60)	0.701			0.94 (0.57–1.54)	0.793		
Time to achieve TTM	0.99 (0.99–1.00)	**<0.001**	1.00 (1.00–1.00)	0.654	1.00 (1.00–1.00)	**<0.001**	0.76 (0.43–1.36)	0.362

*p* < 0.05 are presented in bold. Adjusted for CAG, age, GRACE score (>140), previous PCI, HBP, DM, pulmonary disease, CKD, alcohol, witness, bystander cardiopulmonary resuscitation (CPR), shockable rhythm, time to ROSC (>30), ABGA pH (<7.2), lactate (>7.0) and time to achieve TTM.

**Table 4 jcm-10-00439-t004:** The adjusted odds ratios of early CAG for predicting 6-month good neurologic outcome after out-of-hospital cardiac arrest.

	Total Patients	Early CAG	Good Outcome	OR (95% CI)
GRACE score ≤ 140	151	45	40	1.64 (0.57–4.72)
GRACE score > 140	527	186	78	**2.36 (1.61–3.46)**
Time from collapse to ROSC ≤ 30 min	378	131	89	1.39 (0.89–2.18)
Time from collapse to ROSC > 30 min	300	100	29	**3.89 (2.05–7.38)**
Arterial pH ≥ 7.2	250	93	64	1.29 (0.75–2.23)
Arterial pH < 7.2	403	126	47	**2.06 (1.30–3.26)**
Lactate ≤ 7.0 mmol/L	242	79	53	1.61 (0.92–2.83)
Lactate > 7.0 mmol/L	409	142	60	**2.10 (1.36–3.23)**
ST deviation (+)	470	167	84	**1.91 (1.30–2.80)**
ST deviation (−)	208	64	34	1.46 (0.81–2.63)
Cardiac enzyme elevation (+)	514	193	100	**2.03 (1.41–2.93)**
Cardiac enzyme elevation (−)	162	37	18	1.13 (0.54–2.36)

*p* < 0.05 are presented in bold. OR: odds ratio, CI: confidence interval. Adjusted for CAG, age, GRACE score (>140), previous PCI, HBP, DM, pulmonary disease, CKD, alcohol, witness, bystander CPR, shockable rhythm, time to ROSC (>30), ABGA pH (<7.2), lactate (>7.0) and time to achieve TTM.

## Data Availability

Available upon request.

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
