# Peer review of "Which Out-of-Hospital Cardiac Arrest Patients without ST-Segment Elevation Benefit from Early Coronary Angiography? Results from the Korean Hypothermia Network Prospective Registry"

_jcm, 2021, doi:10.3390/jcm10030439_

Round 1

Reviewer 1 Report

Well written manuscript, good tables and figures, well presented with sensible conclusions 

Author Response

Thank you for your comments. 

Reviewer 2 Report

The authors examined the association between emergency coronary angiography after OHCA using data from a large prospective registry. They report a better neurologic outcome after 6 months in patients undergoing early coronary angiography.

The main strength of this study is the large prospective database.

Broad comments

  1. The authors address an important topic using an observational design. However, there are already data from a randomized trial published 2019 (COACT trial, Lemkes et al, N Engl J Med 380:1397). Thus, in their manuscript the authors should focus on 1) the “added value” of their data to the now available randomized data, and 2) address weaknesses and gaps of available data.
  2. Apparently, the COACT trial was published (2019) after your collection period (2015-2018) but before your analysis and writing-up. You may address this in the methods section and in the limitations together with a statement on whether and how these data changed your focus.
  3. The authors divided their registry data into two categories: “early”, i.e. angiography within 24h and “no-early”, ie angiography > 24h or no angiography at all. I suggest that you split the “no-early” cohort in two cohorts depending on whether they received an angiography: angiography > 24h and no angiography at all. This would allow to assess, whether a coronary angiography in itself has a positive impact on OHCA patients without STEMI (and is a topic not covered by COACT)
  4. The baseline characteristics of your cohorts are not balanced. You could use propensity score matching for more balanced comparisons.

Specific comments

  1. Title: The title should include the notion, that the trial is based on a large registry
  2. Introduction: the introduction should include the randomized controlled COACT trial
  3. Line 84. Difficult to understand. Suggestion. For the present trial, we excluded patients……
  4. Figure 1. If I understood correctly, patients with STEMI were excluded. However, this is not evident from the flow diagram. In my opinion the correct way to display the flow diagram would be to use the box “ECG type unchecked” and write in this box: Arrest with STE (n = 163); arrest with ECG unchecked (n =11); the box “arrest without STE” should than be located in the middle.
  5. Line 116 (and lines 163ff). I did not find results relating to the described multivariate regression. You should either present the results or omit the description of the procedure.  
  6. Table 4, page 9 & figure 2, page 10. The information in the table and the figure is the same. Please omit one.
  7. Line 218. There is a randomized trial (COACT trial) demonstrating that delaying coronary angiography until neurologic recovery is not harmful compared to early angiography.
  8. Line 232. The paragraph starting at line 232 is rather lengthy and difficult to read. Please try to shorten and focus.
  9. Line 251. An important possible explanation between your trial and the randomized COACT trial is selection bias in your cohort. Please add this information  

Author Response

Dear reviewer

Thank you for your comments. We believe the peer-review process has been successful and resulted in a stronger manuscript. Our changes to the manuscript are highlighted and we have included a point-by point discussion regarding the reviewer’s concerns to the document.

Reviewer 3 Report

Thank you for giving me the opportunity to review this study.

In this prospective, multicentric, observational cohort study the authors investigated the association between emergency CAG and outcomes in patients without STE after OHCA. They also identified subgroup of patients with improved outcomes when emergency CAG was performed.

They concluded that CAG performed within 24 hours is associated with good neurologic outcome at 6

months after CA in patients without STE with TTM, identifying patients with high-risk.

The study is well written.

I have some minor correction tips.

Please, specify CAG at first appareance in the abstract;

Delete an “s” in Methods paragraph;

In Table 1, please use the uppercase letter for the column “Early CAG” and “°C” for 33 and 36 target temperature. 

In Table 2, please use the uppercase letter for the column “Early CAG” and specify the unit of measure for the Time to CAG.

Author Response

(The authors gave the same response as above.)
